# Formation Mechanism and Lattice Parameter Investigation for Copper-Substituted Cobalt Ferrites from *Zingiber officinale* and *Elettaria cardamom* Seed Extracts Using Biogenic Route

**DOI:** 10.3390/ma15134374

**Published:** 2022-06-21

**Authors:** Faiqa Barkat, Marina Afzal, Babar Shahzad Khan, Adnan Saeed, Mahwish Bashir, Aiman Mukhtar, Tahir Mehmood, Kaiming Wu

**Affiliations:** 1The State Key Laboratory of Refractories and Metallurgy, Hubei Collaborative Innovation Center for Advanced Steels, International Research Institute for Steel Technology, Wuhan University of Science and Technology, Wuhan 430081, China; tahir10621@yahoo.com (T.M.); wukaiming@wust.edu.cn (K.W.); 2Department of Physics, Government College Women University, Sialkot 51310, Pakistan; faiqabarkat11@gmail.com (F.B.); afzaljanjua84@gmail.com (M.A.); adnan.saeed@gcwus.edu.pk (A.S.); mahwish.bashir@gcwus.edu.pk (M.B.)

**Keywords:** cobalt ferrite, biogenic route, Cu doping, lattice parameter, growth mechanism

## Abstract

Biogenic routes for the synthesis of nanoparticles are environmentally friendly, nontoxic, biocompatible, and cost-effective compared to traditional synthesis methods. In this study, cobalt ferrite was synthesized using *Zingiber officinale* and *Elettaria cardamom* Seed extracts. Effect of copper contents (x = 0.0, 0.3, 0.6 and 0.9) on the plant extracted ${\mathrm{Cu}}_{x}({\mathrm{Co}}_{1-x}{\mathrm{Fe}}_{2}O_{4})$ was investigated by XRD, SEM, EDX, UV-Vis., PL, FE-SEM, FTIR and photocatalytic activity. XRD results revealed that nanoparticles exhibit a cubical spinel structure with an average diameter of 7–45 nm, calculated by the Debye Scherer formula. The value of the lattice parameter decreased from 8.36 Å to 8.08 Å with substitution of copper, which can be attributed to mismatch of ionic radii of Cu^2+^ (0.73 Å) and Co^2+^ (0.74 Å) ions. SEM analysis showed that nanoparticles exhibit a spherical shape (~13 nm diameter) for undoped samples and low Cu concentration, while they changed to a hexagonal structure at higher Cu concentration (x = 0.9) with a diameter ~46 nm and a decreased degree of agglomeration. FE-SEM further confirmed the nanoparticles’ size and shape. EDX analysis confirmed the presence of cobalt, iron, and oxygen without contamination. The optical absorption spectra of UV-vis and PL showed red-shift, which can be accredited to larger crystalline sizes of nanoparticles. FTIR spectra showed two main bands at 410 and 605 cm^−1^, indicating the presence of intrinsic vibrations of the octahedral and tetrahedral complexes, respectively. The photocatalytic activity of Co_0.4_Cu_0.6_ Fe_2_O_4_ nanoparticles was investigated using methylene blue (MB) and methyl orange (MO) dyes under visible light irradiation. The degradation rate (93.39% and 83.15%), regression correlation coefficient (0.9868 and 0.9737) and rate constant (0.04286 and 0.03203 rate·min^−1^) were calculated for MB and MO, respectively. Mechanisms for the formation and photocatalytic activity of Cu-substituted plant-extracted cobalt ferrite were discussed. The Co_0.4_Cu_0.6_ Fe_2_O_4_ nanoferrite was found to be an efficient photocatalyst, and can be exploited for wastewater treatment applications for MB/MO elimination.

## 1. Introduction

Nanotechnology is an interdisciplinary area of research developing new nanoscale structures and examining their properties by altering the particle size, morphology and distribution [1,2,3,4]. Nanosized particles usually exhibit unusual physical and chemical properties that significantly differ from those of bulk equivalents [5,6]. Nanoparticles can be synthesized by chemical and physical approaches, including hydrothermal methods [7], co-precipitation [8,9], sol–gel [10], solid-state reactions [11], microemulsion [12], and electrochemical synthesis [13,14]. However, these methods are of high cost, have lengthy reaction times and high energy requirements and produce non-eco-friendly by products that may hinder large-scale application. Therefore, there is dire need to utilize alternative sustainable routes for the synthesis of nanoparticles that are simple, nontoxic, and cost-effective. Biosynthesis methods include biological sources such as plants, fungus, algae, and microbes as an alternative method to conventional techniques for preparation of large-scale nanocrystalline materials. Among various biological agents, plant extracts are the best candidates because of their vast reserves, ease of access, and their being widely distributed [15].

Spinel ferrite nanoparticles, owing to their unique and interesting optical, magnetic, and antibacterial properties, have gained considerable attention in the biomedical, biosensors, energy storage systems, recording media, data storage, drug delivery, and wastewater treatment fields [16,17,18,19]. Ferrites can be divided in to two spinel structures: normal [20] and inverse [21] spinel. Cobalt ferrite (CoFe_2_O_4_) possesses an Fd_3_m space group with inverse and mixed spinel structure [22,23,24]. Cobalt ferrite is regarded as one of the most interesting metal oxides because of its large magneto crystalline, high coactivity, moderate saturation magnetization, high electrical resistivity, high chemical stability and mechanical strength properties [25,26,27]. These properties of CoFe_2_O_4_ make it suitable for a variety of applications which include high-sensitivity sensors [28], biocompatible magnetic nanoparticles for cancer treatment [29], magnetic resonance imaging [30] and ferro fluid technology [31]. The unique properties of magnetic nanoparticles which are required for biomedical applications are the precise control of particle size, dispersion, antibacterial properties, and biocompatibility. The effect of doping with various transition metals (Mn, Cu, Zn, Ni, etc.) has been investigated by several researchers to improve the physical properties of spinel ferrites [32,33,34,35]. Melo et al. studied Ni-doped cobalt ferrite and showed that the presence of nickel in the cobalt ferrite structure affected the structural, magnetic, and optical properties [34]. The substitution of gadolinium cations in CoFe_2_O_4_ can increase the particle size and alter the saturation magnetization and coercivity values to a lower extent [35]. Margabandhu et al. observed that the copper substitution in cobalt ferrite altered magnetic properties such as coercivity, magnetic retentivity and saturation magnetization [36]. According to Samavati et al., the addition of Cu^2+^ ions in CoFe_2_O_4_ decreases the particle size and level of crystallinity by creating defects and local disorder [33]. Sanpo et al. demonstrated that doping of Cu in cobalt ferrite improves the antibacterial properties and strongly influences the crystal structure, microstructure and particle diameter [37]. Naik et al. reported that substitution of copper in cobalt ferrite leads to decreases in coercivity, saturation magnetization and Curie temperature [38]. The agglomeration of particles occurs so often due to their magnetic nature, which has a detrimental effect on catalysis as it reduces the effective bulk surface area [39]. The properties which are influenced by the addition of Cu in cobalt ferrite include magnetization, diffusion, crystallization, composition, lattice parameter, crystalline size, and phase transformation.

Furthermore, there are few studies on the biosynthesis of CoFe_2_O_4_ nanoparticles by using *Aloe vera*, *Hibiscus rosa-sinensis leaf* and *sesame* Seed extracts [40,41]. Kombaiah et al. synthesized CoFe_2_O_4_ using Okra plant extract as a reducing agent and it showed better structural, optical, and magnetic properties and antimicrobial activities [42]. Gingasu et al. synthesized CoFe_2_O_4_ by using *Hibiscus flower* and leaf extracts as gelling and reducing agents. Their studies showed that cobalt ferrite possesses porous, complex-shaped agglomerates with nano-grained structure with a range of crystallite sizes [43]. Gingasu et al. synthesized CoFe_2_O_4_ nanoparticles using *Zingiber officinale* (ginger) and *Elettaria cardamom* seeds (ECs) extracts, and their results revealed a spinal-type structure with agglomerated, well-defined, faceted crystals [41].

The ginger root extract contains gingerols, shogaols, paradols, and zingerone [44]. The presence of heterocyclic compounds, such as alkanoids, flavoinds, and alkaloids, the active components of dried ginger root, act as reducing, capping and stabilizing agents [45]. Vijaya et al. examined the antibacterial properties of silver nanoparticles (AgNPs) against bacterial pathogens and found that AgNPs capped with *Zingiber officinale* had better microbial activity [45]. Furthermore, ECs extracts are widely used to investigate antibacterial properties. The presence of phytochemicals such as phenols, terpinoids, starch, tannins, proteins, flavonoids and sterols in ECs are capable of reduction and stabilization [18,19]. The presence of an aminoglycosidic antibiotic layer with metal nanoparticles is reported to have an antimicrobial effect on a range of bacterial pathogens [46,47]. Rajan et al. studied the antibacterial activity against *Staphylococcus aureus*, *Escherichia coli* and *Pseudomonas aeruginosa* and cytotoxic activity of HeLa cancer cell lines. The enhanced antibacterial activity and stabilization of gold nanoparticles (AuNPs) were attributed to the presence of capping biomolecules on the surface of AuNps due to the ECs extract [18]. Vinotha et al. synthesized zinc oxide nanoparticles (ZnONPs) by a co-precipitation method using ECs extract. The existence of aromatic compounds were accountable for capping and stabilization of ZnONPs and had enhanced antibiofilm activity against *Enterococcus faecalis, Staphylococcus aureus, Pseudomonas aeruginosa*, and *Proteus vulgaris* [19].

The above studies showed that synthesized nanoparticles from ECs and *Zingiber officinale* have great ability to cap and stabilize the metals. To the best of our knowledge, a single study is present on the synthesis of CoFe_2_O_4_ nanoparticles using ginger root and ECs extract by Gingasu et al. [41], and no data are available on the synthesis of Cu-doped plant-extracted (ECs and ginger root) cobalt ferrite. Hence, little information is available and the formation mechanism for Cu-doped plant extracts is not completely understood. Therefore, the aim of this study to synthesize plant-extracted (ECs and ginger root) cobalt ferrite and investigate the effect of Cu contents in cobalt ferrite. Furthermore, the possible formation mechanism of Cu-doped plant-extracted cobalt ferrite nanoparticles was elaborated.

## 2. Materials and Methods

Copper-substituted cobalt ferrite nanostructures Cu_x_Co_1−x_Fe_2_O_4_ were prepared in two steps that can be found elsewhere [41]. Firstly, ginger root and cardamom seed extracts and nitrates solutions were prepared. Then, these extracts and nitrates solution were mixed to form nanoparticles by a hydrothermal method. Both fresh and dried plant extract can be used as reducing agents to synthesize the nanostructure. However, the dried form is preferred due to it having less water and higher phenolic contents than fresh part [48]. Therefore, plant extracts of ginger roots and cardamom were used in this study. The mechanism of reaction of metal reagents and polyphenols is given elsewhere [9]. The chemicals and solvent used in the study were of the highest purity and analytical grade, purchased from Sigma Aldrich. The ginger roots and cardamom seeds were purchased from a local shop in Sialkot and thoroughly washed with distilled water and dried at room temperature.

### 2.1. Preparation of Ginger Root Extract

Firstly, 10 g of ginger roots was chopped, 50 mL deionized water was added and then the mixture was boiled to obtain a yellowish color, which was completed in 10 min [41]. The extract was cooled down to room temperature, filtered to remove ginger pieces by using Whatman No.1 filter paper and stored in an airtight container for further use. The pH was measured using a 930 Precision pH/Ion Meter, BANTE instruments pH meter (Shanghai, China), and the measured pH was 6.

### 2.2. Preparation of Elettaria cardamom Seed Extract

First, 5 g of *Elettaria cardamom* seed was ground to a fine yellow-brown powder using a modern pestle. The powder was added to 50 mL deionized water under continuous stirring. The solution was boiled for 4 h until its pH became 6. The extract was cooled down to room temperature and filtered using Whatman No.1 filter paper and was stored for further use.

### 2.3. Preparation of Final Solution

Copper-substituted cobalt ferrite (Co_1−x_Cu_x_Fe_2_O_4_) nanostructures were prepared by adding metal nitrates with a ratio of 2Fe^3+^: 1Co^2+^ to the aqueous ginger root and cardamom seed extract under continuous stirring [41]. Then, $1g\;\mathrm{Co}{\left({\mathrm{NO}}_{3}\right)}_{2}\cdot 6H_{2}O$ and $2g\;\mathrm{Fe}{\left({\mathrm{No}}_{3}\right)}_{3}\cdot 9H_{2}O$ were separately dissolved in 25 mL deionized water. The prepared solution of each nitrate was mixed with the ginger root and cardamom seed extracts. To adjust the pH to 12, 2 M solution of NaOH was added drop-wise under continuous stirring for 30 min. The solution was transferred to a Teflon-lined stainless steel autoclave and was placed in a laboratory oven at 160 °C for 12 h. The final product was cooled down to room temperature. At this time, the reaction was completed, resulting in blackish precipitates. These precipitates were centrifuged at 4000 rpm for 30 min, washed several times with deionized water and dried in an oven at 80 °C for 1 h. Four samples were prepared at various copper concentrations (x = 0.0, 0.3, 0.6 and 0.9) by replacing the Co content in plant-extracted cobalt ferrite.

### 2.4. Characterization

Copper-substituted cobalt ferrite nanoparticles were characterized via various techniques. The particles’ crystal phase identification was carried out by an X-ray diffractometer (XRD, Bruker d_2_ Phaser, Berlin, Germany) with Cu-Kα radiation (λ = 1.5406 Å). The external morphology and particles’ sizes were investigated by a scanning electron microscope (SEM) equipped with energy-dispersive spectroscopy (EDX) (FEI Nova NanoSEM 450, Lincoln, NE, USA) with working voltage 10 KV, and 2 theta value ranging from 30° to 70° in a step size of 0.05. The absorption spectra were obtained via UV–visible spectroscopy (UV-Vis., SPECORD 210 Plus, Jena, Germany) with wavelength ranges from 200 to 800 nm. The ASpect UV software was used for recording and analysis of UV/Vis data. The fluorescence analysis was carried out by a Photoluminescence spectrofluorometer (PL, JASCO FP-8200, Madrid, Spain) with a wavelength range from 200 to 800 nm. The full system control of the instrument, data processing, and analysis were achieved with Spectra Manage, a cross-platform spectroscopy software. For UV–Vis and PL analyses, a small amount of powder was dispersed in ethanol and transferred to quartz glass cuvettes for investigation. Fourier Transform Infra-Red spectroscopy (FTIR Bruker Tensor II, GERMANY) in a wavelength range 400 to 2000 cm^−1^ in steps of 4 cm^−1^ was used to examine functional groups of Cu-doped cobalt ferrites. A Field Emission Scanning Electron Microscope (FE-SEM, NOVA Nano SEM 450, USA) was used to study the detailed morphology with operating voltage of 10 KV.

### 2.5. Photocatalytic Activity

The photocatalytic activity (PCA) of Cu-doped cobalt ferrites was evaluated under visible light exposure by degradation of Methylene Blue (MB) and Methyl Orange (MO) dyes in aqueous solutions. In this study, 30 mg of synthesized Co_0.4_Cu_0.6_ Fe_2_O_4_ nanoparticles was added into 50 mL of aqueous solution of MB and MO. The solution was mechanically stirred for 60 min under dark conditions. The degradation of both dyes was noted from the absorption maximum at regular time intervals of 5 min. Then, 3 mL of sample was taken, and nanoparticles were centrifuged to analyze the absorbance of degradation products using a UV–Vis Spectrometer. Degradation rate during absorption was calculated by the following equation [49],
$$Degredation\;Rate\;\left(\%\right)=\frac{A_{o}-A_{t}}{A_{o}}$$
where $A_{o}$ is the initial concentration of dyes and $A_{t}$ is the concentration of dyes at time interval, *t*.

## 3. Results and Discussion

### 3.1. Scanning Electron Microscopy

The morphology and microstructure of copper-substituted plant-extracted cobalt ferrite nanoparticles Co_1−x_Cu_x_Fe_2_O_4_ (0 ≤ x ≤ 1) were observed by SEM coupled with EDX and are presented in Figure 1 and Figure 2, respectively.

Figure 1 shows SEM micrographs for pure CoFe_2_O_4_ synthesized nanoparticles with an average diameter of ~13 nm from the extracts of ginger and cardamom seeds. When the Cu^2+^ is added to the plant-extracted cobalt ferrite (Co_0.7_Cu_0.3_Fe_2_O_4_), the average diameter of nanoparticles increases to ~22 nm. Both samples with x = 0.0 and 0.3 exhibit the agglomeration process that causes formation of large clusters, as shown in Figure 1a,b. The inter-particle interactions caused by van der Waals force and dipole–dipole interaction were mainly responsible for the agglomeration of ferrite nanoparticles [9,50]. With the increase in Cu^2+^ concentration to x = 0.6 and 09, hexagonal-shaped nanoparticles with average diameters ~ 32 nm and ~46 nm, respectively, were formed, and can be seen in Figure 1c,d. The size of the nanoparticles increases with the increase in Cu concentration, while the degree of agglomeration decreases at higher concentrations and the separation of nanoparticles took place.

Figure 2 represents the FE-SEM image of Cu-doped (x = 0.6) cobalt ferrite nanoparticles. The image depicts that nanoparticles have a spherical shape, with an average diameter of ~35 nm, which is in accordance with the SEM and XRD results.

Figure 3a–e represents the elemental mapping of Co_0.4_Cu_0.6_ Fe_2_O_4_ nanoparticles, which indicates the uniform allocation of Cu, Fe, Co, and O elements over the whole region. The elemental composition is in agreement with the EDX spectrum.

### 3.2. Energy-Dispersive X-ray Spectroscopy

The EDX spectra of plant extracted cobalt ferrite and copper substituted cobalt ferrite nanoparticles is presented in Figure 4. The EDX spectra indicate that the nanoparticles are composed of Co, Fe, O, Cu, and C. For pure CoFe_2_O_4_ (x = 0.0), the presence of Fe, Co, and O peaks reveals the purity of the nanoparticles. When the Cu^2+^ is substituted, Cu peaks appeared in the EDX pattern, which confirms that the substitution process was successfully carried out by using the hydrothermal method. The peak of C can be attributed to the carbon tape used as substrate. The results revealed that there is no contamination in the sample. The elemental composition of Cu_x_Co_1−x_Fe_2_O_4_ nanoparticles with x = 0.0, 0.3, 0.6, and 0.9 is presented in Table 1. The compositional analysis for all the samples was found to be close to stoichiometry. Cu-substituted Co ferrites did not deviate from their initial stoichiometry and matched well with the initial degree of Cu substitutions. The results also confirm the increasing amount of Cu composition in the nanoparticles with the increase in Cu content in the sample.

### 3.3. X-ray Diffraction Results

XRD patterns of CuxCo1−xFe2O4 nanoparticles with x = 0.0, 0.3, 0.6, and 0.9 are shown in Figure 4. All samples exhibited a poly-oriented structure, and the peak positions are in coherence with the spinel phase cubic structure. We have not observed any trace of impurity peaks, which confirms that Cu2+ ions have been incorporated into the spinel lattice. The characteristics peaks located at 2θ = 36.14°, 36.98°, 39.40°, 42.86°, 58.84°, 61.89° and 66.62° correspond to (311), (222), (110), (400), (511), (440) and (531) crystal planes, respectively, which is confirmed by international standard card data (JCPDS card No. 22-1086). The addition of Cu to the cobalt ferrite created the new planes that can be clearly seen in Figure 5. This can be attributed to influence of defects or disorders caused by the addition of copper ions in the cobalt ferrite lattice structure. Furthermore, no peaks for metallic Cu/CuO_x_/Cu-Co binary phase are identified which can be based on almost the same atomic radii of Cu^2+^ (0.73 Å) and Co^2+^ (0.74 Å). This proposes that Cu ions simply replace the Co ions in the ferrite crystal lattice without substantially altering the crystal structure of cobalt ferrite [33]. It is worth noting that the substitution of Cu in cobalt ferrite has great impact on size and lattice parameters of cobalt ferrite nanocrystals [33,51]. The investigation of the diameter of nanoparticles has been carried out using the broadening of the XRD peaks. The Debye–Scherer formula was used to calculate the size of copper-substituted cobalt ferrite nanoparticles, and is given as [52]:(1)$$D=\frac{0.9\lambda }{\beta Cos\theta }$$
where β is the full-width half maxima (FWHM) value, D is the crystallite size, λ is the X-ray wavelength (1.54056 Å for Cu-Kα radiation) and θ is the Bragg’s angle. The estimated nanoparticles sizes obtained from the Scherer formula are ~ 7, ~26, ~31 and ~45 nm for x = 0.0, 0.3, 0.6 and 0.9, respectively. The increase in Cu concentration in the solution causes the increase in the size of copper-substituted cobalt ferrite nanoparticles. The average nanoparticles size calculated by the Debye–Scherer formula is in accordance with the SEM results, as depicted in Figure 6a. The lattice constant (a) and the cell volume (V) for the copper-substituted cobalt ferrite nanoparticles as a function of (Cu^2+^) concentration were calculated from the XRD results by using the following relationships (Equation (2)) [53].
(2)$$a={\left[d^{2}\left(h^{2}+k^{2}+l^{2}\right)\right]}^{\frac{1}{2}}\;\mathrm{and}\;V=a^{3}$$

The values of lattice parameter, diameter and volume of plant-extracted pure cobalt ferrite nanoparticles and copper-substituted cobalt ferrite nanoparticles are tabulated in Table 2. The value of the lattice parameter for pure CoFe_2_O_4_ is a=8.360 Å and is decreased from 8.313 Å to 8.084 Å with the substitution of Cu in cobalt ferrite, which agrees with the reported results [33]. The lattice parameter values have an almost linear dependence on the Cu contents, as shown in Figure 6b, which obeys Vegard’s law [54]. The decreasing trend in the unit cell parameter can be ascribed to the replacement of larger ionic radius of Co^2+^ (0.74 Å) by the smaller ionic radii of Cu^2+^ (0.73 Å) in the host system [33,51].

### 3.4. UV–Visible Spectroscopy Results

UV–visible spectral analysis has been widely used to characterize semiconductor nanoparticles. The absorption spectra of Co ferrite and Cu-doped cobalt ferrite nanoparticles in the UV light region is illustrated in Figure 7. The sample possessed an absorption band in the whole range and exhibited good absorption in the light region (300–400 nm). The absorption at 300 nm is assigned to the characterization absorption band of CoFe_2_O_4_ nanoparticles. On substituting Cu in cobalt ferrite, the absorption band is shifted towards longer wavelengths, as shown in Figure 7. The fundamental absorption which corresponds to electron excitation from the valence band to conduction band can be used to determine the value of the optical band gap of synthesized Co_1−x_Cu_x_Fe_2_O_4_ nanoparticles. The band gap can be obtained from a linear extrapolation of absorbance edge to the wavelength axis. The estimated band gap values of Co_1−x_Cu_x_Fe_2_O_4_
with (x=0.0, 0.3, 0.6 and 0.9) nanostructure was found to be 4.13–3.44 eV. The band gap energy decreases with increasing Cu content, which may be associated with various parameters including the crystalline size, structural parameter, carrier concentration, presence of very small amounts of impurities which are detectable by XRD techniques and lattice strains [55]. A change in the absorption intensity and a shift in the band are attributed to a shift in ion concentration at the sites [56].

### 3.5. Photoluminescence Results

Photoluminescence spectra are used to study the luminescence properties and for the determination of band gap energies. Figure 8 shows the PL spectra of Cu-doped cobalt ferrite (Co_1−x_Cu_x_Fe_2_O_4_) nanoparticles with (x = 0.0, 0.3, 0.6, 0.9). Un-doped cobalt ferrite at x = 0.0 and doped cobalt ferrite with concentration x = 0.3 show an emission peak at 461 nm, while the emission peaks for x = 0.6 and x = 0.9 are found at 490 nm. The estimated band gap energies are in the range of (2.68–2.53 eV). It can clearly be seen that by increasing the concentration of Cu, band gap energy decreases, which shifts the emission spectra towards a longer wavelength.

### 3.6. FTIR Spectroscopy

Figure 9 illustrates the Cu-substituted Co_1−x_Cu_x_Fe_2_O_4_) (x = 0.3, 0.6 and 0.9) nanostructures in the range of 400–2000 cm^−1^. Two main bands observed at 410 and 605 cm^−1^ are related to octahedral and tetrahedral metal oxides’ positions, respectively [57]. The presence of these bands confirms the phase purity, as observed in XRD data. Metal oxide bands are observed in same position in all three samples as Cu ions have almost the same radius as Co ions. The substitution of Cu with Co does not alter the sample crystal structure.

However, the intensity of the bands increases with substitution of Cu^2+^, which strengthens the ferrite’s structure [58]. The difference in band position due to Fe^3+^ and O^2−^ for octahedral and tetrahedral structures is due to their difference in restoring force [59]. The C–O vibration is correlated with the IR band found around 1030 cm^–1^ [60]. A symmetric stretching mode of vibrations of C–O was observed near 1256 cm^−1^. The band observed at 1530 cm^–1^ is due to the symmetrical stretching of the carboxylate [61]. Formation of carboxylates is due to excessive energy during synthesis provided by organics.

### 3.7. Photocatalytic Activity

UV–Vis spectra showing changes in dyes’ concentration and kinetics of degradation with Co_0.4_Cu_0.6_ Fe_2_O_4_ of MB and MO dyes are presented in Figure 10 and Figure 11, respectively. The irradiation time was 60 min in intervals of 5 min. UV–vis spectra revealed that the intensity of absorbance peaks for MB and MO dyes with Co_0.4_Cu_0.6_ Fe_2_O_4_ nanoparticles decreases with time, as shown in Figure 10a and Figure 11a, respectively. Figure 10b and Figure 11b illustrate the % degradation rate vs. irradiated time for MB and MO in presence of Co_0.4_Cu_0.6_ Fe_2_O_4_ nanoparticles. The observed degradation rate for MB was 93.39% and it was 83.15% for MO. The kinetics of degradation were calculated by plotting the graph between $\ln\left(\frac{A_{o}}{A_{t}}\right)\;$ and irradiated time, as shown in Figure 10c and Figure 11c for MB and MO dyes, respectively. The linear fitting of the plot indicates that the degradation of both dyes obeys the kinetics of the pseudo first-order effect [49,62]. The regression correlation coefficient ($R^{2}$) factor has a value of 0.9868, and rate constant (K) is calculated from the slope of graph and has value of 0.04286 rate·min^−1^ for MB dye. For MO, $R^{2}$ is 0.9737 and K = 0.03203 rate·min^−1^. The above results revealed that Co_0.4_Cu_0.6_ Fe_2_O_4_ nanoferrite is an efficient photocatalyst and can be exploited for wastewater treatment applications for MB/MO elimination.

## 4. Discussion

### 4.1. Mechanism for the Formation of Cobalt Ferrite Nanoparticles

The green synthesis of nanoparticles has some advantages over the conventional methods as they limit the use and production of toxic, inorganic chemicals and can be carried out at ambient conditions while still preserving the quality of the nanostructures with a relatively fast production rate. For nanoparticle synthesis facilitated by plant extracts, the extract is mixed with metal precursor solutions under different reaction conditions [63]. The exact mechanisms for the biological synthesis of nanoparticles have not yet been extensively studied, as each autotroph contains different compounds responsible for the reaction. However, the general process consist of an activation phase in which the nucleation process begins following the metal ion reduction. Then, by the process of Ostwald ripening, tiny particles combine spontaneously to grow nanoparticles. At the end, the termination phase defines the final shape of the nanoparticles [64,65]. In the present study, metal ions Co^2+^ and Fe^3+^ undergo reduction processes in the presence of phytochemicals/heterocyclic compounds to form neutral metal ions Co and Fe to start the nucleation process. The aromatic compounds present in ECs and ginger root extracts are responsible for capping and stabilizing the nanoparticles. Finally, cobalt ferrite nanoparticles growth commences, where they are formed as the precipitate within the mixture. The overall growth mechanism is presented in Figure 12. The reactions for synthesis of cobalt ferrites in nitrate solutions can be summarized in three stages: (i) formation of metal hydroxide, (ii) formation of metal complexes and (iii) formation of spinel ferrites (AB_2_O_4_) [34]. The general formula for the formation of metal hydroxide is given in Equation (3),
(3)$$M{\left({\mathrm{NO}}_{3}\right)}_{y}\cdot \left(a\right)H_{2}O+\left(b\right)\mathrm{NaOH}\to M{\left(\mathrm{OH}\right)}_{y,\mathrm{sol}.}+\left(b\right){\left[\mathrm{NaOH}\right]}_{3,\mathrm{aqu}.}+\left(c\right){\left[H_{2}O\right]}_{\mathrm{liq}.}$$
where M stands for metal in metal nitrate, y depends on reagent, and a, b, and c are the equation-balancing constants. The reagents used in this study are $\mathrm{Co}{\left({\mathrm{NO}}_{3}\right)}_{2}\cdot 6H_{2}O$ and $2\mathrm{Fe}{\left({\mathrm{No}}_{3}\right)}_{3}\cdot 9H_{2}O$, so, reactions can be written as [34].
(4)$$\left\{\begin{array}{c} Co{\left(NO_{3}\right)}_{2}\cdot 6H_{2}O+2NaOH\to Co{\left(\mathrm{OH}\right)}_{2,\mathrm{sol}.}+2{\left(\mathrm{NaOH}\right)}_{3,\mathrm{aqu}.}+6{\left(H_{2}O\right)}_{\mathrm{liq}.}\\ 2Fe{\left(NO_{3}\right)}_{3}\cdot 9H_{2}O+6NaOH\to 2Fe{\left(\mathrm{OH}\right)}_{3,\mathrm{sol}.}+6{\left(\mathrm{NaOH}\right)}_{3,\mathrm{aqu}.}+18{\left(H_{2}O\right)}_{\mathrm{liq}.} \end{array}\right\}$$

In Equation (4), Co^2+^ and Fe^3+^ react with NaOH to form $M{\left(\mathrm{OH}\right)}_{y,\mathrm{sol}.},$ and further these hydroxides react with sodium hydroxide to form metal complexes. The general reaction is presented in Equation (5),
(5)$$M{\left(\mathrm{OH}\right)}_{y}+\left(b\right)\left[\mathrm{NaOH}\right]\to M{\left(\mathrm{OH}\right)}_{2y}^{y^{-}}+\left(b\right){\mathrm{Na}}^{+}$$

The formed complexes of Co and Fe are mixed and reacted together to form the cobalt spinel ferrite (Equation (6)),
(6)$$\mathrm{Co}{\left(\mathrm{OH}\right)}_{4}^{2^{-}}+2\mathrm{Fe}{\left(\mathrm{OH}\right)}_{6}^{3^{-}}+8{\mathrm{Na}}^{+}\to {\mathrm{CoFe}}_{2}O_{3}+8\mathrm{NaOH}+4H_{2}O$$

The overall reaction for the Cu substituted cobalt ferrite is presented in Equation (7) [34],
(7)$$\left(1-x\right)\mathrm{Co}{\left({\mathrm{No}}_{3}\right)}_{2\;}\cdot 6H_{2}O+2\mathrm{Fe}{\left({\mathrm{No}}_{3}\right)}_{3}\cdot 9H_{2}O+\mathrm{xCu}{\left({\mathrm{No}}_{3\;}\right)}_{2}\cdot 6H_{2}O\to {\mathrm{Co}}_{1-x}{\mathrm{Cu}}_{x}{\mathrm{Fe}}_{2}O_{4}+8{\mathrm{NaNo}}_{3\;}+28H_{2}O$$

### 4.2. Cu Concentration Effect on Lattice Parameter of Cobalt Ferrite Nanoparticles

Usually, in a solid solution of ferrites, a linear change in lattice constant with concentration of the components is observed [33,51]. It is observed in this study that CoFe_2_O_4_ nanoparticle size increases while the lattice parameter decreases with increasing Cu, in good agreement with the literature [66]. These changes can be explained as follows.

To understand various physical properties of AB_2_O_4_, knowledge of the valence state and distribution of cations over tetrahedral (A-) and octahedral (B-) sites of spinel lattices is essential. Cation arrangements are not unique in spinel ferrites. For an equilibrium structure, each spinel compound possesses at least three degrees of freedom: (i) degree of cation inversion (x), (ii) in addition to general position of anions, oxygen positional parameter (u) for complete description, and (iii) lattice constant (a).

Generally, any two cations A and B may have different sizes and occupy tetrahedral and octahedral sites, respectively. For (AB_2_O_4_), single A occupies the tetrahedral site and B_2_ occupies two equivalent octahedral sites, referred to as Normal (x→0) [20]. However, Barth and Posnjak (1932) suggested that the alternative distribution of spinel ferrite is also possible, i.e., tetrahedral site occupancy by B and octahedral site occupancy by A and B, mentioned as inverse (x→1) [21]. x is the degree of cation inversion, which is the fraction of tetrahedral sites occupied by B ions. The variations in x result in a corresponding adjustment of two structural parameters u and a [20], which has dependence on various factors: temperature, electrostatic contribution to the lattice energy, pressure, crystal-field effects, cationic radii and cationic charge and some of the principal factors that influence cation inversion [20].

The parameter u mostly varies with the radius ratio of A- and B-site cations and is related to the movement of O ions in the spinel ferrites system, giving an idea about the degree of distortion of anions (O^2−^). The discrepancy in u alters the bond length between sites A and B to accommodate the cations in a unit cell. In general, most spinel ferrites exhibit a slightly larger value of u than the ideal value (u_0_ = 0.375; taking origin at A site), so, O^2−^ ions move away along the <111> direction from the cations to enlarge tetrahedral sites at the expense of octahedral sites. The value of u increases slightly with the increase in Cu concentration, and can be attributed to the migration of Fe^3+^ ions from B sites to A sites [20]. The tetrahedral lattice sites are slightly smaller in respect to metal ions, which results in shrinking of octahedral sites to settle the A sites [20,66,67,68].

The decrease in lattice parameter in this study can be explained by the mismatch of ionic radii. Cu^2+^ ions have smaller radii (0.73 Å) than Co^2+^ ions (0.74 Å) and preferentially occupy B sites over A sites [69]. Cu^2+^ ions replace Co^2+^ ionic radii on B sites and migrate towards the smaller A sites when B sites become Cu-rich. Additionally, the increase in Cu contents in the solution increases the value of u, which forces O^2−^ ions to move away from cations to enlarge the A site by reducing the B site and to decrease the mean ionic radii of the B site. The shrinkage of B sites reduces the bond length, thereby decreasing the lattice parameter and overall volume of the unit cell by substitution of Cu^2+^ content in place of Co^2+^ ions in the samples. Furthermore, the increased crystalline nature with increasing size of ${\mathrm{Cu}}_{x}\left({\mathrm{Co}}_{1-x}{\mathrm{Fe}}_{2}O_{4}\right)$ nanoparticles by substituting Cu results in higher structural stability by lowering the overall strain energy [20]. From SEM and XRD analysis, it was found that nanoparticle size of copper-doped cobalt ferrites increases with the increase of Cu concentration. Therefore, it may be considered that overall strain energy is decreased to form stable nanoparticles.

The optical absorption spectra of UV–vis and PL results showed red-shift, and large size nanoparticles are confirmed by XRD and SEM analysis for ${\mathrm{Cu}}_{x}\left({\mathrm{Co}}_{1-x}{\mathrm{Fe}}_{2}O_{4}\right)$ nanoparticles with an increase in Cu concentration. The shift in band gap can be accredited to two effects; (i) blue-shift due to quantum confinement and (ii) stress-induced effects owing to large crystalline size responsible for red-shift [70,71]. Therefore, red-shift in ${\mathrm{Cu}}_{x}\left({\mathrm{Co}}_{1-x}{\mathrm{Fe}}_{2}O_{4}\right)$ nanoparticles due to the Cu concentrations can be attributed to the large-sized nanoparticles.

### 4.3. Mechanism for Dyes’ Degradation

Cu-doped cobalt ferrite nanoparticles and MB/MO solutions were irradiated by visible light, and electrons (e^−^) in the valence band (VB) were excited to the conduction band (CB) creating holes (h^+^) in the VB [72]. The surface water molecules reacted with holes to produce hydroxyl radicals (*OH) and superoxide anion radicals (*O_2_) were formed due to the dissolves oxygen molecules present with electrons in CB [73]. Generally, e^−^ and h^+^ recombined quickly to reduce the photocatalytic activity of the catalyst [74]. However, it was observed that Cu substitution in cobalt ferrites enhanced the photodegradation of cobalt ferrites [72,74]. The reason is that dopants not only tune the optical bandgap of the doped photocatalyst, as examined in this study, but also facilitate minimization of e^−^/h^+^ via serving as a strong agent for charges [75,76]. It is well-known that photocatalytic activity is a surface-dependant process; therefore, previous studies showed that, specifically in the spinel structure, octahedral sites are surface-exposed, which enhances photocatalytic performance of the ferrite system. As discussed earlier, Cu ions replaced the Co ions on octahedral sites resulting in decreased overall volume of the unit cell. In our case, this could be a possible reason for the very efficient photocatalytic activity of Co_0.4_Cu_0.6_Fe_2_O_4_ for elimination of MB and MO with catalytic performances of 93.39% and 83.15%, respectively, under irradiation for 60 min.

## 5. Conclusions

A biogenic route was developed to synthesize cobalt ferrite nanoparticles using *Zingiber officinale* and *Elettaria cardamom* seeds extracts. The heterocyclic compounds/phytochemicals were capable of reduction and stabilization of ferrite nanoparticles. The structural, morphological, and optical properties of Cu-substituted cobalt ferrite nanoparticles were investigated. The XRD results confirmed the cubic spinel structure for ${\mathrm{Cu}}_{x}({\mathrm{Co}}_{1-x}{\mathrm{Fe}}_{2}O_{4})$ nanoparticles. It was observed that crystalline size increases and lattice constant decreases by increasing copper content in plant-extracted cobalt ferrite. The spherical shape for undoped and low-Cu-concentration samples changed to a hexagonal shape for high-Cu-substituted cobalt ferrite with a decrease in the degree of agglomeration. The optical band gap of samples was calculated in the visible region which was confirmed by UV–Vis and PL analyses. The band gap energy value was decreased at Cu concentrations (x=0.6 and 0.9) in ${\mathrm{Cu}}_{x}({\mathrm{Co}}_{1-x}{\mathrm{Fe}}_{2}O_{4})$, which can be attributed to the larger crystalline size of nanoparticles. The smaller Cu^2+^ ions’ radius (0.73 Å) compared to that of Co^2+^ ions (0.74 Å) was responsible for shrinkage of octahedral sites, resulting in a decrease in lattice parameter. The efficient photocatalytic activity of Co_0.4_Cu_0.6_Fe_2_O_4_ for elimination of MB and MO with catalytic performances of 93.39% and 83.15%, respectively, under irradiation for 60 min, was observed.

## Figures and Tables

**Figure 1 materials-15-04374-f001:**
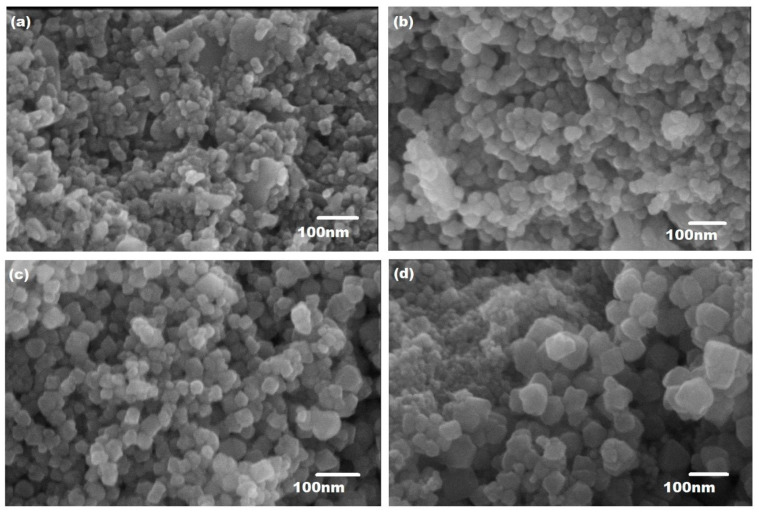
SEM images of Cu^2+^-doped plant-extracted CoFe_2_O_4_ nanoparticles: (**a**) pure CoFe_2_O_4_, (**b**) Co_0.7_Cu_0.3_ Fe_2_O_4_, (**c**) Co_0.4_Cu_0.6_ Fe_2_O_4_ and (**d**) Co_0.1_Cu_0.9_ Fe_2_O_4_.

**Figure 2 materials-15-04374-f002:**
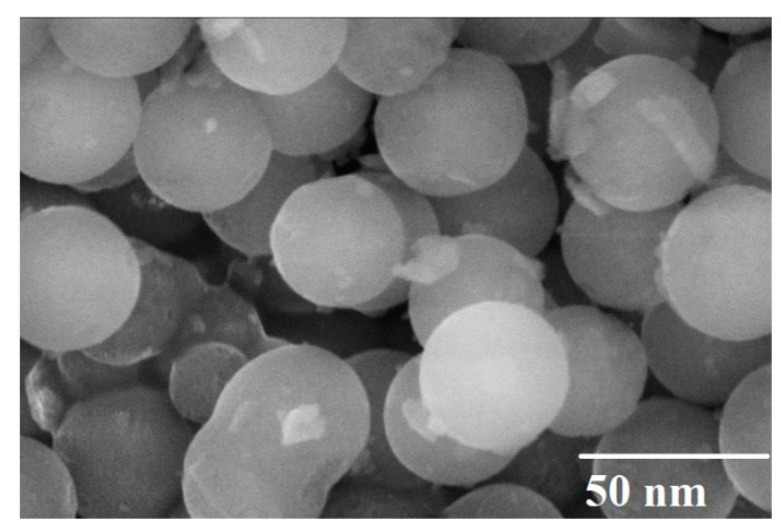
FE-SEM image of Cu-doped, plant-extracted CoFe_2_O_4_ nanoparticles at x = 0.6.

**Figure 3 materials-15-04374-f003:**
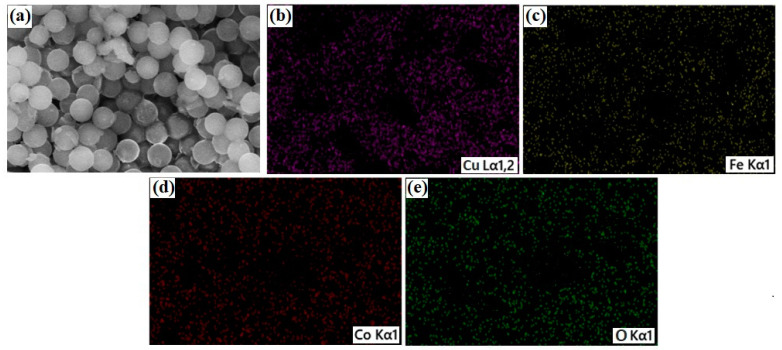
Elemental mapping of Co_0.4_Cu_0.6_ Fe_2_O_4_ nanoparticles (**a**) FE-SEM, (**b**) Cu, (**c**) Fe, (**d**) Co and (**e**) O.

**Figure 4 materials-15-04374-f004:**
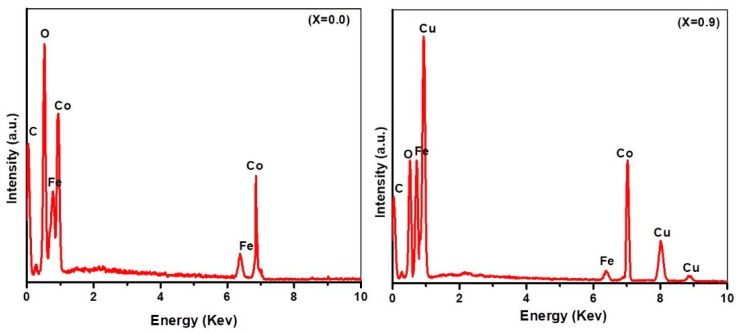
EDX spectra of synthesized, plant-extracted, pure cobalt ferrite nanoparticles at x = 0.0 and copper-substituted cobalt ferrite nanoparticles (Co_0.1_Cu_0.9_ Fe_2_O_4_) at x = 0.9.

**Figure 5 materials-15-04374-f005:**
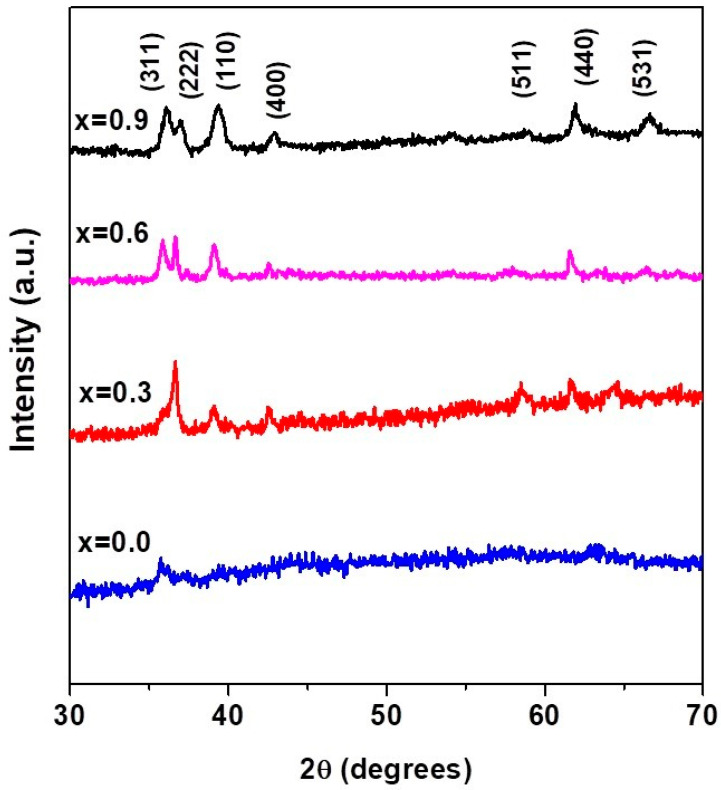
XRD pattern of copper-substituted, plant-extracted cobalt ferrite nanoparticles Co_1−x_Cu_x_Fe_2_O_4_ (0 ≤ x ≤ 1): at x = 0.0; pure CoFe_2_O_4_, at x = 0.3; Co_0.7_Cu_0.3_ Fe_2_O_4,_ at x = 0.6; Co_0.4_Cu_0.6_ Fe_2_O_4_ and at x = 0.9; Co_0.1_Cu_0.9_ Fe_2_O_4_.

**Figure 6 materials-15-04374-f006:**
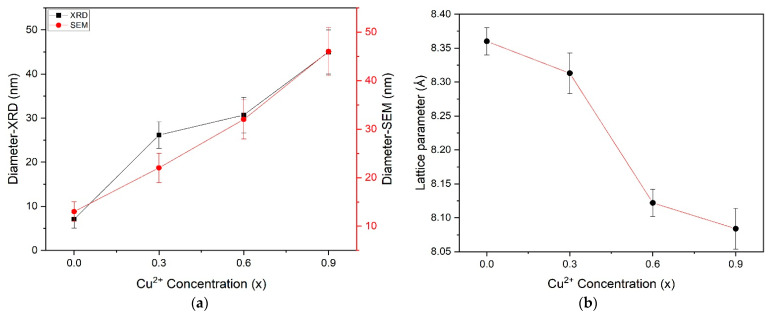
Graphical representation of lattice parameter (**a**) and diameter calculated by Debye–Scherer formula and SEM (**b**) of plant-extracted cobalt ferrite nanoparticles Co_1−x_Cu_x_Fe_2_O_4_.

**Figure 7 materials-15-04374-f007:**
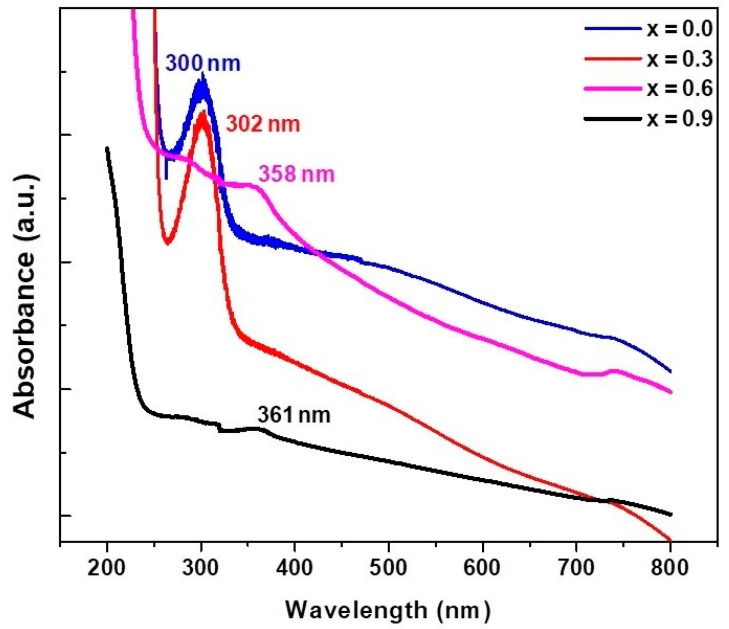
UV–visible spectra copper-substituted, plant-extracted cobalt ferrite nanoparticles Co_1−x_Cu_x_Fe_2_O_4_ (0 ≤ x ≤ 1): at x = 0.0; pure CoFe_2_O_4_, at x = 0.3; Co_0.7_Cu_0.3_ Fe_2_O_4_, at x = 0.6; Co_0.4_Cu_0.6_ Fe_2_O_4_ and at x = 0.9; Co_0.1_Cu_0.9_ Fe_2_O_4_.

**Figure 8 materials-15-04374-f008:**
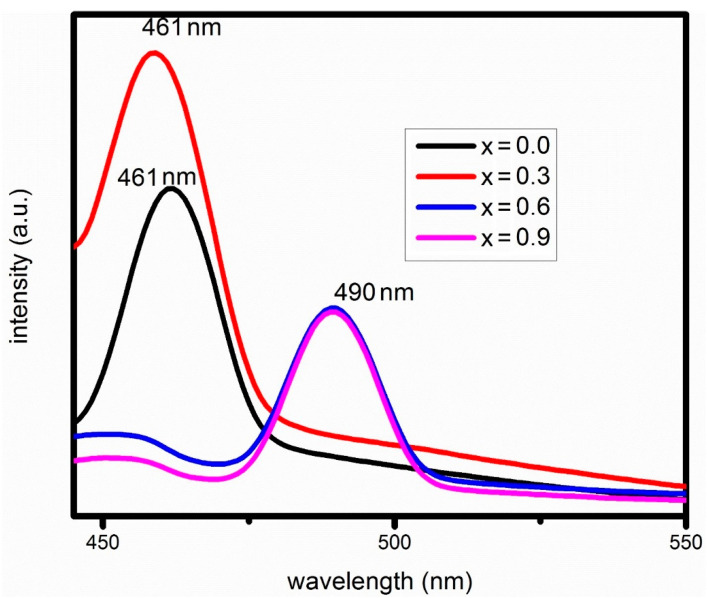
PL spectra of Cu-doped cobalt ferrite (Co_1−x_Cu_x_Fe_2_O_4_) nanoparticles with (x = 0.0, 0.3, 0.6, 0.9).

**Figure 9 materials-15-04374-f009:**
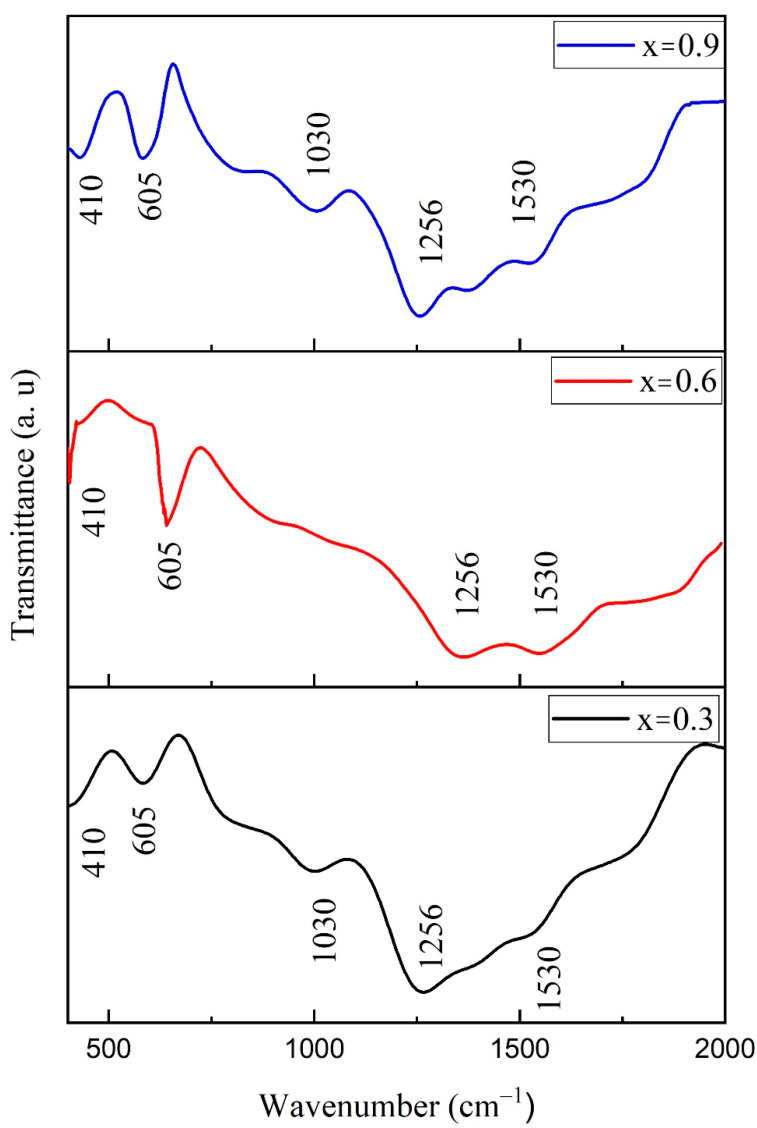
FTIR spectra of Cu-doped cobalt ferrite (Co_1−x_Cu_x_Fe_2_O_4_) nanoparticles with (x = 0.3, 0.6, 0.9).

**Figure 10 materials-15-04374-f010:**
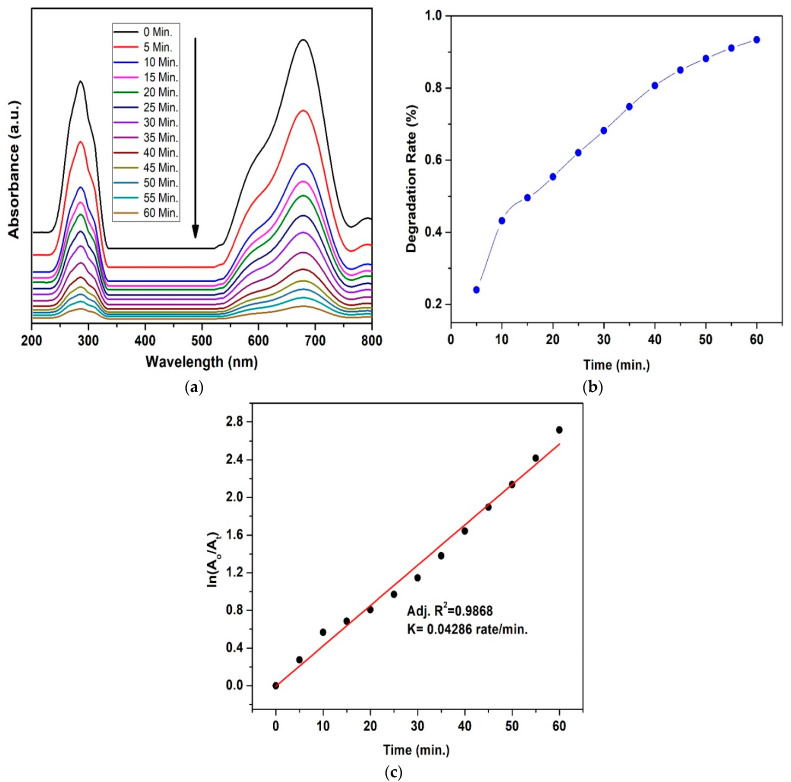
Photocatalytic activity of MB in presence of Co_0.4_Cu_0.6_ Fe_2_O_4_ nanoparticles irradiated by visible light for 60 min: (**a**) UV–Vis. Spectra, (**b**) Degradation rate in % and (**c**) kinetics of degradation.

**Figure 11 materials-15-04374-f011:**
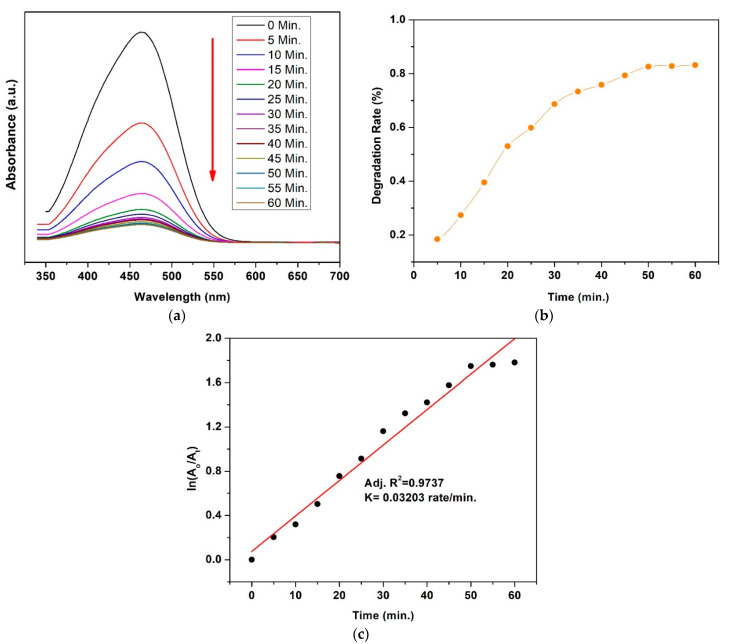
Photocatalytic activity of MO in presence of Co_0.4_Cu_0.6_ Fe_2_O_4_ nanoparticles irradiated by visible light for 60 min: (**a**) UV–Vis. Spectra, (**b**) Degradation rate in % and (**c**) kinetics of degradation.

**Figure 12 materials-15-04374-f012:**
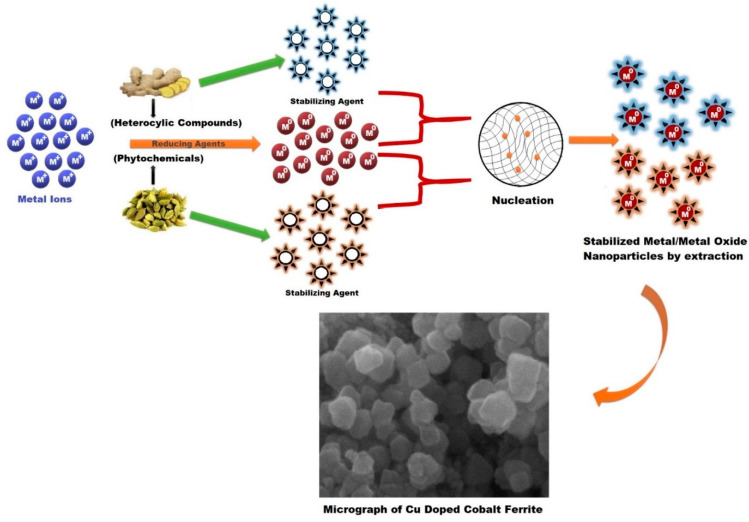
Schematic representation of growth mechanism of Cu_x_Co_1−x_Fe_2_O_4_: metal ions from precursors undergo reduction process in presence of phytochemicals/heterocyclic compounds. The process is initiated through nucleation of metallic substances; thereafter, nanoparticle growth commences.

**Table 1 materials-15-04374-t001:** The elemental composition of plant-extracted pure cobalt ferrite nanoparticles and copper-substituted cobalt ferrite nanoparticles.

Ferrite Composition (Cu_x_Co_1−x_Fe_2_O_4_)	Value of x (0≤x≤1)	Element wt.%				
		C	Co	Cu	Fe	O
CoFe_2_O_4_	0.0	2.37	63.06	-----	11.06	23.51
Co_0.7_Cu_0.3_ Fe_2_O_4_	0.3	3.66	21.25	41.29	8.63	25.17
Co_0.4_Cu_0.6_ Fe_2_O_4_	0.6	3.65	11.06	54.14	8.04	23.11
Co_0.1_Cu_0.9_ Fe_2_O_4_	0.9	3.55	3.08	65.14	5.63	22.60

**Table 2 materials-15-04374-t002:** The values of lattice parameter, diameter and volume of plant-extracted pure cobalt ferrite nanoparticles and copper-substituted cobalt ferrite nanoparticles.

Ferrite Composition (Cu_x_Co_1−x_Fe_2_O_4_)	x (0≤x≤1)	a (Å)	$d_{hkl}$	~Diameter (nm)		$V\;{\left[\left(Å\right)\right]}^{3}$
				Scherer	SEM	
CoFe_2_O_4_	0.0	8.360	2.5095	7.06	13	584.2771
Co_0.7_Cu_0.3_ Fe_2_O_4_	0.3	8.313	2.4532	26.15	22	574.4779
Co_0.4_Cu_0.6_ Fe_2_O_4_	0.6	8.122	2.4461	30.69	32	535.783
Co_0.1_Cu_0.9_ Fe_2_O_4_	0.9	8.084	2.4257	45	46	528.2979

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
