# Peer review of "Formation Mechanism and Lattice Parameter Investigation for Copper-Substituted Cobalt Ferrites from Zingiber officinale and Elettaria cardamom Seed Extracts Using Biogenic Route"

_materials, 2022, doi:10.3390/ma15134374_

Round 1

Reviewer 1 Report

1.       Methods must be improved. Why two stabilizing agents (Zingiber officinale and Elettaria Cardamom seeds extracts) have been used? Isn’t it possible to synthesize this nanostructure by using single reducing agent?

2.       TEM images are needed to understand proper shape and size of the particles.

3.       Elemental mapping of single particle is needed to justify the production of composite (copper substituted cobalt ferrites) nanoparticles—EDAX is not enough.

4.       What is the application of the particles? -Include at least one.

Reviewer 2 Report

The manuscript materials-1753698: Formation mechanism and lattice parameter investigation for copper substituted cobalt ferrites from Zingiber officinale and Elettaria Cardamom seeds extracts using biogenic route by Barkat et al. described the synthesis of copper-substituted cobalt ferrites using Zingiber officinale and Elettaria Cardamom seeds extracts. The authors evaluated the effects of Cu content on the product by XRD, SEM, EDX, UV-Vis spectroscopy, and photoluminescence spectroscopy. Based on a previous study (Gingasu et al.) that synthesized cobalt ferrites from Zingiber officinale and Elettaria Cardamom extracts, this study continued with the Cu substitution. Overall, the study has novelty and contribution to the field. However, the manuscript should be considerly revised according to the following comments to be published in the Materials.

1. This study repeated the synthesis of cobalt ferrites from a previous study (Gingasu et al.). The authors should compare the results of two studies.

2. The main focus of this study is the effects of Cu content. Therefore, these effects should be intensively discussed and compared with previous studies using Cu substitution.

3. “Nanoparticles synthesized through this biogenic route can potentially be used in various biomedical applications” (Abstract). This point is essential and should be expanded in the discussion.

4. Should Zingiber officinale and Elettaria Cardamom seeds extracts be characterized before the preparation of the final solution?

5. References should be cited for the methods described in the method section.

6. In Gingasu et al. paper, the FTIR data were used to investigate the formation of CoFe2O4 spinel structure. Do the authors perform such FTIR analysis? If yes, please include and discuss. If not, please explain the reasons?

7. Page 7, lines 1-4 from the bottom: all the methods should be described in detail (sample preparation, operation parameters, instrument model and country of origin).

8. Different styles were used, such as Margabandhu et al., C.C. Naik et al., (2018), Gingasu et al, and  GINGAÅžU et.al. Please keep them consistent.

9. Most of the references (>75%) were published in 1957–2016. If possible, please replace some of them with updated ones.

10. Page 4, lines 9-11: the authors should rewrite the unclear sentence.

11. Page 21, line 14: does x = 0.3 represent a higher Cu concentration?

12. Data in Figure 4 should be presented as means ± SDs if possible.

13. The authors should carefully check and correct grammar errors and typos throughout the manuscript.

Round 2

Reviewer 1 Report

1. Authors are suggested to improve methods ( add some subheads viz. characterizations and dye degradation).

2. Elemental mapping is required to understand the composition of the synthesized nanostructures. FTIR only revealed the surface functional groups.

Reviewer 2 Report

The manuscript was appropriately revised and can be accepted for publication without further revision. 

Author Response

The reviewer has accepted the revision without any further change.